# Dopant-Dependent Electrical and Biological Functionality of PEDOT in Bioelectronics

**DOI:** 10.3390/polym13121948

**Published:** 2021-06-11

**Authors:** Małgorzata Skorupa, Daria Więcławska, Dominika Czerwińska-Główka, Magdalena Skonieczna, Katarzyna Krukiewicz

**Affiliations:** 1Department of Physical Chemistry and Technology of Polymers, Silesian University of Technology, 44-100 Gliwice, Poland; malgorzata.skorupa@polsl.pl (M.S.); daria.m.wieclawska@gmail.com (D.W.); dominika.czerwinska-glowka@polsl.pl (D.C.-G.); 2Biotechnology Centre, Silesian University of Technology, 44-100 Gliwice, Poland; magdalena.skonieczna@polsl.pl; 3Department of Systems Biology and Engineering, Faculty of Automatic Control, Electronics and Computer Science, Silesian University of Technology, 44-100 Gliwice, Poland

**Keywords:** bioelectronics, conducting polymer, doping, neural interface, PEDOT

## Abstract

The aspiration to interact living cells with electronics challenges researchers to develop materials working at the interface of these two distinct environments. A successful interfacing coating should exhibit both biocompatibility and desired functionality of a bio-integrated device. Taking into account biodiversity, the tissue interface should be fine-tuned to the specific requirements of the bioelectronic systems. In this study, we pointed to electrochemical doping of conducting polymers as a strategy enabling the efficient manufacturing of interfacing platforms, in which features could be easily adjusted. Consequently, we fabricated conducting films based on a poly(3,4-ethylenedioxythiophene) (PEDOT) matrix, with properties modulated through doping with selected ions: PSS^−^ (poly(styrene sulfonate)), ClO_4_^−^ (perchlorate), and PF_6_^−^ (hexafluorophosphate). Striving to extend the knowledge on the relationships governing the dopant effect on PEDOT films, the samples were characterized in terms of their chemical, morphological, and electrochemical properties. To investigate the impact of the materials on attachment and growth of cells, rat neuroblastoma B35 cells were cultured on their surface and analyzed using scanning electron microscopy and biological assays. Eventually, it was shown that through the choice of a dopant and doping conditions, PEDOT-based materials can be efficiently tuned with diversified physicochemical properties. Therefore, our results proved electrochemical doping of PEDOT as a valuable strategy facilitating the development of promising tissue interfacing materials with characteristics tailored as required.

## 1. Introduction

Electroactive interfacing of living cells constitutes the core of bioelectronic applications whose strategy is based on a cell–substrate communication. By this approach, striving to enable a controlled flow of information between the recipients, it is possible to interact electronics with a biological matter [1]. This concept is primarily employed in biomedical engineering purposes such as biosensors, tissue engineering, or neuroprosthetics [2], to allow for either sensing, stimulation, or control of biological communities on functional surfaces. The crucial role in a possibly seamless interconnection of those two distinct worlds of biotic and abiotic nature is played by the interfacing platforms, which need to fit several criteria. The materials of interest should be characterized by an ability to provide an electrical trigger to cells and should exhibit low impedance, suitable surface architecture, and biocompatibility [1]. Additionally, the growing variety of bioelectronic devices has been designed specifically for various cellular requirements, which in turn demands new advanced interfacing platforms adjusted to those particular needs.

So far, numerous formulations have been studied for such purposes, however, there is still an ongoing effort towards achieving increased agility in the manufacturing of high-performance engineered surfaces. A great promise to create such tailorable interfacing platforms was shown by conducting polymers, which are believed to serve as next generation engineering materials particularly beneficial in the development of supercapacitors [3] and anticorrosion coatings [4]. In particular, poly(3,4-ethylenedioxythiophene) (PEDOT) draws considerable attention in providing electroactive coatings due to its high conductivity, high stability, and good biocompatibility, already shown for endothelial, epithelial, fibroblasts, macrophage, and human neural cell lines [5,6]. Recently, different forms of PEDOT were introduced as excellent biointerfaces, including organic platforms for neural stimulation and recording (PEDOT:Nafion) [7], hard tissue scaffolds (composite of PEDOT:PSS, gelatin and bioactive glass) [8], physiological strain sensors (composite of PEDOT:PSS functionalized CNTs and poly(glycerol sebacate urethane)) [9], and platforms for tissue engineering and organoid approaches (PEDOT:PSS crosslinked via glycidoxypropyltrimethoxysilane) [10].

However, another substantial reason for its attractiveness for bioelectronics, in addition to the palette of advantageous properties, is found in the ability to tune its physicochemical characteristics. Current strategies of modulating the behavior of PEDOT comprise the fabrication of composite structures with other conducting polymers, such as polyaniline [11], decoration of PEDOT surface with metallic particles [12], as well as electrosynthesis in the presence of various doping ions [13]. Essentially, doping leads to the modification of the electronic structure of polymers and influences their conductivity. This occurs by incorporating doping agents, which introduce charge carriers into the polymer chain and neutralize the unstable oxidized or reduced polymer backbone. Importantly, it is known that the dopant identity, namely its mobility and size, influences the features of conducting polymers and consequently has an impact on the interactions of cell–substrate systems [14]. Virtually, even trace amounts of introduced ions translate into markedly modified film properties, such as conductivity, ability to store electrical charge, biocompatibility, and morphology [13].

The selection of a doping agent appears to be a key factor in the design of diversified bioelectronic interfaces. However, there is still a disagreement regarding the understanding of actual relationships between the dopant character and the end-performance of a resultant coating. Baek et al. [15] demonstrated that doping with small ions results in the fabrication of PEDOT films, exhibiting increased surface roughness, lower impedance, and higher charge transfer capacity. The findings reported by Valle et al. [16] supported the relationship between the choice of a doping ion and polymer morphology, but were in contradiction to the previous reports when it comes to the conductivity: Valle claimed enhanced conductivity upon increasing ion size. Based on the above background and reported discrepancies, we considered it important to identify the exact correlation of dopant-film features and to leverage fluent fabrication of modulable polymeric interfaces.

In this work, we investigated three PEDOT-based materials obtained by incorporating different anionic dopants during the polymerization of EDOT. To meet the requirements of agile production, synthesis by electrochemical polymerization was employed, which permitted an efficient, easily controlled process yielding tailored products. As doping agents, PSS^−^ (poly(styrenesulfonate)), ClO_4_^−^ (perchlorate), and PF_6_^−^ (hexafluorophosphate) were selected. To discuss how these different counterions affected morphological, biological, and electrical parameters of a PEDOT matrix, doped polymer films were investigated in terms of their conductivity, capacitance, chemical composition, surface morphology, and biocompatibility.

## 2. Materials and Methods

### 2.1. Reagents

3,4-ethylenedioxythiophene (EDOT), poly(sodium 4-styrenesulfonate) (NaPSS, average M_w_ = 1,000,000 g/mol), and tetrabutylammonium hexafluorophosphate (nBu_4_NPF_6_) were obtained from Sigma Aldrich (Saint Louis, MO, USA). nBu_4_NPF_6_ was vacuum dried before use. Lithium perchlorate (LiClO_4_) and potassium chloride (KCl) were obtained from Acros Organics. Deionized water (Millipore quality) and acetonitrile (ACN, HPLC grade, Sigma Aldrich) were used as solvents.

### 2.2. Electrochemical Polymerization

The electrochemical polymerization of EDOT was performed by means of a CH Instruments, Inc. type 660C potentiostat (Austin, TX, USA) by a cyclic voltammetry (CV) scanning. A standard three-electrode setup was used, comprising a Ag/AgCl (3 M KCl) reference electrode (ET073, EDAQ, Denistone East, Australia), a platinum plate counter-electrode (1 cm^2^) (Mennica Polska, Warsaw, Poland), and a platinum plate working electrode (0.283 cm^2^) (Mennica Polska, Warsaw, Poland). The polymerization of EDOT (10 mM) was carried out in three different systems, consisting of different electrolyte salts and solvents, namely: NaPSS/water, nBu_4_NPF_6_/ACN and LiClO_4_/water. The electrolyte concentration was 0.1 M for both nBu_4_NPF_6_ and LiClO_4_, and for NaPSS it was calculated concerning a single structural unit (0.1 M). In all cases, the scan rate was equal to 100 mV/s and the polymerization was conducted over 30 CV cycles.

### 2.3. Chemical and Morphological Characterization

Morphological characterization was performed with a scanning electron microscope (SEM) Phenom ProX (Phenom-World BV, Eindhoven, The Netherlands) operating at 15 kV. Spectrochemical properties of investigated materials were analyzed using Raman spectroscopy (Renishaw inVia, Renishaw, Wotton under Edge, UK, 633 nm excitation laser) in the spectral range of 400–1600 cm^−1^.

### 2.4. Electrochemical Characterization

The electrochemical studies were performed by means of a potentiostat equipped with a three-electrode system mentioned earlier, in 0.1 M KCl aqueous solution. Cyclic voltammograms were collected over the potential range from −0.4 V to 0.7 V (vs. Ag/AgCl) for five CV cycles with a scan rate of 100 mV/s. Charge storage capacity (CSC) was calculated by integration of the area under the CV curve. Polymer voltammetric capacitance (C_cv_) was obtained by dividing the integrated area under theCV curve by two to obtain mean cathodic and anodic capacitance [17]. Electrochemical impedance spectroscopy (EIS) measurements were performed using CH Instruments, Inc. type 660C potentiostat (Austin, TX, USA), in 0.1 M KCl over frequencies ranging from 100 mHz to 10 kHz, with an AC amplitude of 10 mV and a DC potential of 0 V (vs. Ag/AgCl). The recorded spectra were fitted into an equivalent circuit by means of EIS Spectrum Analyser 1.0 [18] with the application of a Powell algorithm.

### 2.5. In Vitro Biological Characterization

Biological characterization was performed based on a rat neuroblastoma cell line B35 (CRL-2754™, ATCC, Manassas, VA, USA). Cells were grown in a Dulbecco’s Modified Eagle Medium/Nutrient Mixture F-12 (DMEM/F12, Sigma-Aldrich) supplemented with 10% fetal bovine serum (FBS, Gibco, Long Island, NY, USA) and gentamicin (40 mg/mL, Krka Poland Sp. z o.o., Warsaw, Poland) at 37 °C in constant 80% humidity atmospheres and 5% carbon dioxide concentration (Heracell™ 150i, Thermo Scientific, Waltham, MA, USA). The cells were trypsinized with a 0.25% trypsin-EDTA solution (Sigma-Aldrich) in PBS, then neutralized by the addition of an equal amount of culture medium. Cells were seeded at 2 × 10^5^ cells/well in 2 mL medium on 12-well plates (Biologix, Shandong, China) and were cultured for 48 h. Subsequently, after the addition of 1 mL of 0.25% trypsin-EDTA solution (Sigma-Aldrich) to each well, resulting cell suspensions were centrifuged at 1500 rpm for 3 min.

The cytotoxicity of reported materials was determined based on the ability of viable cells to reduce tetrazolium dye MTT (3-[4,5-dimethylthiazol-2-yl]- 2,5-diphenyltetrazolium bromide) to insoluble formazan. Consequently, 50 µL of MTT solution (0.05 mg/mL in phenol red and FBS free DMEM-F12; PAA) was added to samples collected from 12-well plates. After 1–2 h in a CO_2_ incubator, MTT solution was removed and the resulting formazan crystals were dissolved in 400 µL acidic isopropanol. Absorbance measurements at 570 nm were performed using a multi-well plate reader SYNERGY4 (BioTek Instruments, New York, NY, USA).

The cell cycle of B35 neuroblastoma cells on PEDOT surfaces was analyzed by a flow cytometry (BD FACSAria^TM^ III, Becton, Dickinson and Company, Franklin Lakes, NJ, USA). Cells, after centrifugation, were washed with 500 µL PBS and centrifuged again. Then, the cells were stained with 250 µL of hypotonic buffer (comprised from PI 100 µg/mL in PBS; 5 mg/L of citric acid; 1:9 Triton-X solution; RNase 100 µg/mL in PBS from Sigma, Poznan, Poland) and DNA levels were assessed by fluorescence measurements at a PE configuration (547 nm excitation laser line; emission: 585 nm).

The percentage of apoptotic and necrotic cells after 48 h of exposure on different samples was measured using FITC Annexin V Apoptosis Detection Kit with PI (Bio Legend, San Diego, CA, USA). Fifty microliters of Annexin V Binding Buffer, 2.5 μL of FITC-conjugated Annexin V antibody, and 10 μL propidium iodide (100 μg/mL) were added on each surface. Subsequently, the samples were vortexed gently and incubated in dark for 20 min at a room temperature (25 °C). Before measurements, 250 μL of Annexin V Binding Buffer was added to each tube. Flow cytometric analysis enabled to determine the fluorescence of PI (necrotic cells) at the configuration for PE channel, and for AnnexinV FITC-conjugated antibody (apoptotic cells) at FITC channel configuration (488 nm excitation laser line; emission: LP mirror 503, BP filter 530/30).

The morphology of cells cultured on investigated surfaces was examined using SEM microscopy (Phenom ProX). B35 neuroblastoma cells were fixed using 3% glutaraldehyde (Fisher BioReagents, Fair Lawn, NJ, USA) for 24 h, then washed three times with deionized water. Samples were dehydrated by a 10 min immersion in the solutions of ethanol (Acros Organics, Morris Plains, NJ, USA) with increasing concentrations (30%, 50%, 70%, 80%, 90%, 95%, and 99.8%), and dried for 24 h at 50 °C. Then, samples were sputter-coated with a gold layer (20 min, 20 mA; Q150R Quorum Technologies, Laughton, UK), and SEM images were taken at 10 kV.

## 3. Results

### 3.1. Electrochemical Polymerization

As demonstrated in several previous studies [19], by choosing a doping agent, it is possible to affect the polymerization potential of the monomer, the efficiency of polymer formation, and its electrochemical characteristics. On that account, for each investigated system, the polymerization potential ranges were optimized separately. In all cases, the starting potential was chosen as −0.8 V (vs. Ag/AgCl), which was associated with the lower limit of a water window for a bare platinum electrode. The oxidative potentials were selected based on the position of a polymerization peak (anodic peak responsible for irreversible oxidation of the monomer [20]) and its vicinity: onset potential (E_o_), the half-peak potential (E_1/2_), the peak potential (E_p_), and the overoxidation potential (E_oo_), all approximated to ±0.1 V (Figure 1A–C). Upper potential limits for the polymerization of EDOT were optimized in terms of best electrical performance expressed by high CSC and low impedance at 1 kHz (Figure 1D–F). Eventually, the anodic potentials of 1.8 V, 1.2 V, and 1.2 V were chosen for nBu_4_NPF_6_/ACN, LiClO_4_/H_2_O, and NaPSS/H_2_O, respectively, and the corresponding polymerization curves of EDOT were plotted in Figure 1G–I.

In each case, a typical, gradual increase in current density was observed in consecutive CV scans and was associated with the growth of a conducting polymer layer [21]. As easily observed from the magnitude of achieved currents, the efficiency of the polymerization process was dependent on the choice of a dopant/solvent system [21]. Among all investigated systems, the most prominent increase in current intensity was noted for PEDOT/PF_6_. This finding is consistent with literature reports stating that, generally, conducting polymer films formed in organic solvents are more conductive than those formed in aqueous solutions [22,23]. This is a result of a greater conjugation length of the polymer and more concise contact with the electrode surface [24] and may arise from different solubility of PEDOT oligomers in different solvents at the initial stages of polymerization [25]. The oligomers precipitate more readily in ACN, attaching to the electrode surface and forming a high number of nucleation centers [25,26]. Another possible explanation for decreased efficiency of polymerization in aqueous solutions is attributed to the nucleophilic character of water molecules, which may lead to a competing reaction of water decomposition by interacting with initially formed polarons [25].

Unlike other conjugated polymers, the finite potential window of polythiophenes is difficult to observe because of the close proximity of the oxidative doping and oxidative degradation potentials [27]. Literature reports suggest that polythiophenes undergo overoxidation above the potentials of 1.45–1.55 V vs. SCE (1.48–1.58 V vs. Ag/AgCl [28]), causing an irreversible damage to the conjugated system and impairing its electrical conductivity [29,30]. However, PEDOT polymerized in nBu_4_NPF_6_/ACN solution with a higher-end potential of 1.8 V (vs. Ag/AgCl) exhibited superior electrochemical properties among all tested potential ranges (Figure 1E). In this work, contrary to experiments demonstrating overoxidation of PEDOT [27,31], the electropolymerization was carried out in the presence of an excess of EDOT in the reaction solution. Therefore, at the anodic potential of 1.8 V (vs. Ag/AgCl), the oxidation of monomer might occur preferentially over the polymer degradation, leading to the formation of a polymer, exhibiting superior electrochemical properties. The deposition potentials optimized for the remaining two systems were in line with previously reported experiments, namely 1.2 V for both LiClO_4_/H_2_O [19] and NaPSS/H_2_O [32].

### 3.2. Surface Characterization

It is well known that the polymerization conditions, such as potential range and the choice of a solvent and doping ion, greatly influence not only the conductivity of the matrix but also its surface morphology [19]. Moreover, these characteristics are directly related, as an increase in surface area translates into lowering of impedance [33]. This was also observed for PEDOT grown under investigated conditions (Figure 2A–C). PEDOT/PF_6_ exhibited a highly developed, porous surface with micro- and nano-sized pores, resembling a sponge composed of interconnected chains. The surface of PEDOT/ClO_4_ was more compact, displaying distinct, globular grains of varying sizes (3–11 µm in diameter with an average of 5.88 ± 1.85 µm). Similar observations for PF_6_^-^ and ClO_4_^−^ doped PEDOT matrices were previously made by Culebras et al. [34]. On the other hand, the surface of PEDOT/PSS was identified as nonuniform, with numerous smaller and larger cavities revealing platinum surface. Also, when PEDOT/PSS surface was scanned by a highly energetic electron probe (15 kV), the polymer film underwent degradation and multiple cracks were observed with the naked eye. Suffering from poor durability, PEDOT/PSS was found to be the least stable polymer compared to other investigated samples, which did not exhibit degradation after SEM exposure.

Generally, our results correlated with earlier research stating that the porosity is more enhanced with the decrease in the size of the doping ion [15]. It was confirmed that grainy, developed film architectures are typical for small dopants, such as ClO_4_^−^ or PF_6_^−^ ions [19,34], whereas more compact structures are typical for large polymeric dopants like PSS [35]. Although ion size difference is negligible between PF_6_^−^ (2.42 Å [36]) and ClO_4_^−^ (2.25 Å [36]), the surface is more porous upon doping with PF_6_^−^ (S_a_ = 2.07 µm) than ClO_4_^−^ ions (S_a_ = 1.79 µm). This should be, however, associated with the effect of solvent, because according to multiple previous reports, the choice of solvent has a stronger influence on the morphology than the electrolyte [26,37]. For example, Seki et al. [25] indicated that water molecules may form hydrogen bonds with the conducting polymer, thus hindering its formation.

Surface morphology directly affects cellular adhesion. Rough topologies of conducting polymers were found to provide more suitable interfaces for cell attachment, compared to smooth metallic surfaces, conventionally applied as neural interfaces [21,38]. Nevertheless, other studies reported that the roughness factor is not decisive itself, but a crucial role is played by a correlation of cell size with the size of topographical details [34,39]. Comparable sizes occur to be favorable because a higher area of contact provides stronger and more stable attachment of cells or biofilm to the substrate [40]. Consequently, depending on the required application, surfaces should be individually examined for each cell type. For typical human neurons, the diameter of cell bodies is 10–50 µm and the diameter of axons lies within the range of 1–25 µm, whereas the neurites can measure as small as a fraction of a micrometer [41]. The somas of B35 neuroblastoma cells used in this study are approximately 10 µm in diameter. While the polymer architecture might not be ideally compatible with the cell body size, the cavities formed between the distinct polymer structures could offer better integration with axonal connections than a completely flat platinum surface.

The effect of dopant nature on structural properties of PEDOT samples was investigated by Raman spectroscopy (Figure 2D). In general, the acquired spectra of PEDOT doped with different counterions were similar to each other, and to those reported previously [34,42,43], indicating that the chemical nature of PEDOT was maintained. Particularly, the most intensive bands located at 1515 cm^−1^, 1426 cm^−1^, and 1366 cm^−1^ corresponded to asymmetric C_α_=C_β_ stretching, symmetric C_α_=C_β_ stretching, and C_β_–C_β_ stretching, respectively. The band at 1266 cm^−1^ originated from stretching modes of C_α_–C_α_ (inter-ring); C–O–C deformation appears at 1096 cm^−1^. The modes at 990 cm^−1^, 575 cm^−1^, and 442 cm^−1^ were ascribed to the oxyethylene ring deformation. The band at 698 cm^−1^ was related to symmetric deformation modes of C–S–C.

### 3.3. Electrochemical Characterization

To compare electrochemical properties of PEDOT matrices, their CV curves were collected in a monomer-free 0.1 M KCl aqueous solution in the same potential range corresponding to the redox activity of PEDOT [19]. CV curves of all investigated coatings (Figure 3A) were of a rectangular shape, suggesting their largely capacitive character [1]. The most developed shape of a CV curve was exhibited by PEDOT/PF_6_, followed by PEDOT/ClO_4_ and PEDOT/PSS, respectively. All three polymers largely outperformed a bare platinum electrode, for which the CV curve resembled a flat line. To quantitatively compare the ability of PEDOT matrices to store electric charge, a charge storage capacity (CSC) was calculated from corresponding CV curves. CSC is a parameter applied frequently to compare the performance of bioelectronic interfaces [44]. In general, high values of CSC are desired, since they allow for using lower potentials to obtain a high current response. As presented in Figure 3B, the highest value of CSC (80.1 ± 6.3 mC/cm^2^) was noted for PEDOT/PF_6_, outperforming greatly both PEDOT/ClO_4_ (CSC of 36.6 ± 1.8 mC/cm^2^) and PEDOT/PSS (CSC of 18.4 ± 1.23 mC/cm^2^), as well as a bare Pt electrode (1.5 ± 0.2 mC/cm^2^).

Aurian-Blajeni et al. [33] showed that CSC is controlled by surface morphology. High effective surface area (surface to area ratio) allows for a larger part of the coating surface to be directly in contact with the solution, therefore enabling an increased charge transfer between the solution and the electrode coating. This correlation was also apparent in this work, with highly porous PEDOT/PF_6_ exhibiting the greatest value of CSC. Interestingly, it is PEDOT/PSS with a typical CSC of 10–26 mC/cm^2^ [45,46,47] that is a formulation often used in various bioelectronic applications [48]. Although a CSC value determined in this study was in the typical range for PEDOT/PSS (18.4 ± 1.3 mC/cm^2^), still, this material was found as the least capacitive among investigated PEDOT matrices. The relatively poor capacitance of PEDOT/PSS was related to the polymeric nature of PSS and, therefore, increased spacing between PEDOT chains [15], making the interchain hopping of electrons more difficult than in the case of small doping ions.

Impedance is a crucial factor in bioelectronic considerations, as low impedance is necessary for enhancing charge transfer between the electrode and surrounding tissue. Low impedance profile is also responsible for maintaining a high signal to noise ratio in neural recording [49]. Usually, impedance behavior of electrodes is expressed as a magnitude of impedance modulus (|Z|). A relevant benchmark value used for comparing materials for bioelectronic applications, especially for neural electrodes, is |Z| measured at the frequency of 1 kHz [50,51], which is the frequency associated with a neuron firing. Bode plots presenting the impedance module as a function of frequency (Figure 4A) revealed a similar level of |Z| at 1 kHz for all investigated polymer samples (Figure 4C), with a small prevalence of PEDOT/PF_6_ (113.8 ± 0.5 Ω). The analysis of phase angle vs. frequency plots (Figure 4B) demonstrated notable variations in the phase angle, particularly at low frequencies, suggesting distinct capacitance of the studied PEDOT films.

To assess the resistance and capacitance of PEDOT films, EIS experimental data were fitted with the use of a Powell algorithm to an equivalent circuit model. As the most relevant, the modified Randles circuit was used, consisting of solution resistance (R_s_), charge transfer resistance (R_ct_), double-layer capacitance (C_dl_), Warburg diffusion element (Z_w_), and polymer bulk redox capacitance (C_d_). The presence of the latter was suggested by the shape of a Nyquist plot, and a necessity to simulate an element accounting for the phase angle shift from 45° to 90°. Such a model was previously proposed by Danielsson et al. [52] for describing the behavior of PEDOT electrosynthesized in ionic liquids and was used by Kim et al. [53] to characterize PEDOT doped with polydopamine. EIS data exhibited sufficient goodness of fit (χ^2^ values below 0.001), proving a suitable choice of an equivalent circuit. The fitted spectra, together with experimental data, were presented in Figure 4A,B.

The electronic capacitive character of the polymer coating is represented by a capacitor element C_d_ in the equivalent circuit. The highest capacitance concerning the electrode surface was obtained for PEDOT/PF_6_ (47.6 ± 7.1 mF/cm^2^), which was three times greater than C_d_ of PEDOT/ClO_4_ (15.0 ± 0.8 mF/cm^2^) and six times greater than C_d_ of PEDOT/PSS (7.0 ± 0.7 mF/cm^2^). For comparison, the aerial capacitance of a PEDOT/PSS coating, prepared from chemically synthesized dispersions, is typically of approx. 1 mF/cm^2^ [54,55]. The capacitance value achieved by PEDOT/PF_6_ should be considered as impressive, since the aerial (interface) capacitance of supercapacitors determined basing on CV curves recorded at the same scan rate (100 mV/s) is usually of a similar order of magnitude (e.g., 75 mF/cm^2^ as noted for NiCo_2_O_4_ thin films [56]). Because of the lack of distinctive redox signals in CV curves of all PEDOT formulations, they should be considered as intrinsic pseudocapacitors, i.e., those in which the charge is mainly stored within an electric double layer rather than through faradaic redox reactions [57].

Critically, the polymerization solvent was found to possess a more pronounced impact on the capacitance at the polymer surface than a dopant, which was associated with a more developed morphology and more surface available for direct contact with the electrolyte. It could be hypothesized that the observed variation in the capacitance of PEDOT-based materials is related to the changes in the conformation of a conducting polymer matrix occurring as a result of doping with ions of different size and structure, in different environments. It is expected that a key role is played by a chain separation. Recent studies have shown that by modulating chain separation in PEDOT, it is possible to change the properties of the polymer. For instance, an incorporation of Nafion into PEDOT:PSS improved power conversion efficiency of polymer solar cells [58], and a simple treatment of PEDOT:PSS by the organic solvent greatly enhanced the thermoelectric power factor of the polymer [59]. Therefore, it is possible that the changes in chain separation could induce the formation of interfacial polarization, similarly as it happens in conducting polymer composites [60].

By the juxtaposition of the capacitance obtained from equivalent circuit fitting with values calculated from CV curves (Figure 4D), a good correlation between C_d_ and CSC was evident, confirming the correct fitting of EIS data. Because of an inconclusive fitting of the high frequency region, the values of resistance were compared as a sum of R_s_ and R_ct_, presented as R, corresponding to the high frequency resistance, and being close to the resistance measured at 1 kHz (Figure 4E). It turned out that the resistance of all the polymer coatings in this study was virtually identical, showing that the choice of dopant had a little influence on the polymer resistance while greatly affecting its capacitive properties.

### 3.4. Biological Characterization

Although PEDOT is generally recognized as a biocompatible material [48], the presence of different dopants could influence its interactions with tissues. Therefore, a standard colorimetric assay (MTT) was used to compare the viability of cells cultured on different PEDOT formulations with the viability of cells cultured on a bare Pt surface and in an empty well (Figure 5A). A rat B35 neuroblastoma cell line was used to assess the biocompatibility of investigated materials, since it is an easily transfected, cultured *cell* model of central nervous system neurons. After 48 h of incubation, a reduced percentage of viable cells (85 ± 3%) was noted on a surface of bare platinum, which is generally regarded as a biocompatible material [61]. Interestingly, when the surface of bare Pt was coated with PEDOT/ClO_4_, cell viability was slightly improved (89 ± 7%). Even though perchlorates are known to possess the anti-thyrotoxic activity and interfere with iodide uptake [62], it seems that their interactions with a PEDOT matrix were strong enough to prevent their elution in concentrations from being harmful to cells.

According to the ISO 10993-5:2009 standard [63], a material can be regarded as non-cytotoxic when the cell survivability rate exceeds 70% concerning the control group. Therefore, among three investigated PEDOT formulations, only PEDOT/ClO_4_ can be considered biocompatible. The viability of cells cultured on PEDOT/PF_6_ was equal to 69 ± 4%, which placed it at the boundary of biocompatibility, whereas the presence of PEDOT/PSS resulted in the reduction of cellular metabolic activity to 48 ± 5%, rendering it nonviable. According to previous cytotoxicity studies on doped PEDOT films [15], NaPSS might induce cell inhibition, since it was found to be more toxic than LiClO_4_ at the concentration of 1 mg/mL.

Apart from cell viability, the choice of a dopant could also affect the cell cycle. Accordingly, the cytometric evaluation of the cell cycle was performed, and the cells were divided into four phases, namely sub-G_1_ (inactive cells with damaged DNA), G_0_/G_1_ (cells at a resting point), S (DNA synthesis), and G_2_/M (cells undergoing mitosis) (Figure 5B). For all experimental groups, the largest portion of cells remained in the G_0_/G_1_ phase. Elevated fraction of B35 cells arrested in this phase, compared to the control group, should be associated with the inhibition of cell development. However, none of the electrode surfaces increased the portion of cells arrested in the G_0_/G_1_ phase, and therefore had no blocking effect on the cellular development [64]. Being responsible for DNA synthesis and mitosis, S and G_2_/M phases indicate the proliferation of cells. The most significant part of B35 cells remaining in S and G_2_/M phases was observed for PEDOT/ClO_4_ and Pt. Therefore, these surfaces could be considered as supporting cell multiplication. In the case of PEDOT/PSS, a larger percentage of cells was damaged than in the proliferative phases, indicating its destructive effect on the cell culture. This observation was supported by the highest percentage of B35 cells in the sub-G_1_ phase (29% for PEDOT/PSS), showing that PEDOT/PSS significantly induced cell death by either apoptosis or necrosis, and this effect was more severe than in the case of PEDOT/ClO_4_ (7% of cells in sub-G_1_ phase) and PEDOT/PF_6_ (15% of cells in sub-G_1_ phase). Interestingly, the difference between platinum and control sample was insignificant at this stage.

To get a better insight into the origin of cell damage, an apoptosis assay was performed (Figure 5C). The control population was characterized by a high percentage of normal cells (94%), which proved proper execution of the experiment [65]. Most cell death occurred by necrosis (6%), and only less than 1% entered the physiological apoptosis phase. It is supposed that this cell death distribution might be just a characteristic of this specific cell line, as indicated in a previous study [66]. A high percentage of healthy cells was also noted for PEDOT/ClO_4_, with a similar level of necrotic cells as in the control group. Some part of the cells (9%) were in the state of early or late apoptosis. Apoptotic death, however, was exceedingly induced on the surface of PEDOT/PF_6_ film (31%), and this could be a result of the presence of trace amounts of ACN. A similar amount of healthy cells as for PEDOT/PF_6_ (58%) was noted for PEDOT/PSS (60%). However, together with a platinum surface, PEDOT/PSS exhibited the highest number of dead cells as a result of necrosis—a pathological, mechanical cell death. It proved them both to be the most deteriorating to cell survivability, even though the cell phase distribution of cells on Pt surface did not indicate such behavior. This discrepancy could result from the high variability of interactions between cells and Pt surface, which made it difficult to accurately predict the response of tissue towards implantable devices.

Apart from supporting the viability of cells, an interface material should also enhance their outgrowth. Therefore, SEM imaging was used to analyze the topography of cells cultured on investigated PEDOT formulations and a Pt control (Figure 6). It was observed that B35 cells formed a developed, highly branched axon network, particularly on the surface of PEDOT/ClO_4_. The cells were found to form the interconnections with a rough polymer layer, confirming that the polymer was well-fitted to the size of cells. In the case of PEDOT/PF_6_, the integration of neural cells into the foam-like structure of a polymer matrix was even more pronounced. Due to the corrugated surface of PEDOT interface, the axons intertwined with the polymeric chains. PEDOT/PSS, which was flatter than the other polymers, did not exhibit increased integration with B35 cells. Besides, large clumps of cells were observed on the surface of PEDOT/PSS, which could originate from cell disintegration [67]. Although the smooth surface of platinum provided good adherence of B35 cells, the formation of more sophisticated superficial interactions was not observed.

## 4. Conclusions

In this work, three PEDOT-based materials were fabricated by electrochemical polymerization of EDOT in the presence of different counterions, namely PSS^−^ (poly(styrenesulfonate)), ClO_4_^−^ (perchlorate), and PF_6_^−^ (hexafluorophosphate). Despite the common polymeric matrix, the materials varied substantially in terms of capacitive, morphological, and biological properties. Our results not only distinctly demonstrated the powerful impact of a dopant nature on the overall film characteristics, but also served as an incentive for detecting the potential of alternative dopants, other than commonly applied PSS, in the design of bioelectronic devices. It was shown that both PEDOT/PF_6_ and PEDOT/ClO_4_ outperformed PEDOT/PSS in terms of electrical and biological properties. Impressive electrochemical capacitance of PEDOT/PF_6_ and superior biocompatibility profile of PEDOT/ClO_4_ left behind the results collected for either PEDOT/PSS or a bare platinum electrode. Although PEDOT, and more precisely PEDOT/PSS, is already a well-established player in many research areas including bio-integrated applications, our study restated the applicability of PEDOT in bioelectronics. Eventually, the material end-performance was a synergetic effect of the pivotal nature of PEDOT, modulated by the effect of dopant, and supported by solvent and synthesis parameters selection. Accordingly, we draw attention to doping as a prospective approach that enables fabricating PEDOT substrates with desired functionalities. This strategy could be invaluably facilitated by completing the understanding of the dopant-polymer intertwined relationships. Although our research supplements this knowledge, further studies should be undertaken, taking into account also other alternative counterions. Establishing a library of such assignments would pave the way toward the agile development of conductive platforms imparted with features meticulously tailored to specific interfacing applications.

## Figures and Tables

**Figure 1 polymers-13-01948-f001:**
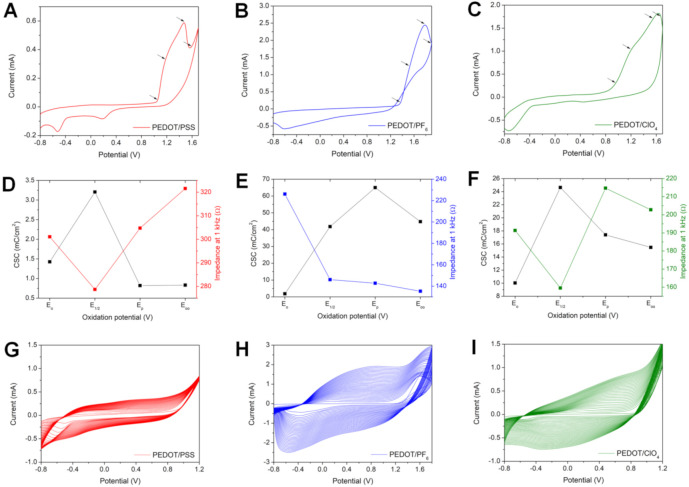
Optimization of electropolymerization conditions for each doping system (PEDOT/PSS, PEDOT/PF_6_, and PEDOT/ClO_4_). (**A**–**C**) CV curves showing the selection of oxidative potentials based on the position of a polymerization peak, where E_o_ is the onset potential, E_1/2_ is the half-peak potential, E_p_ is the peak potential, and E_oo_ is the overoxidation potential. (**D**–**F**) The variation of charge storage capacity (CSC) and impedance modulus at 1 kHz with the choice of an oxidative potential. (**G**–**I**) CV curves showing the electropolymerization of EDOT under optimized conditions.

**Figure 2 polymers-13-01948-f002:**
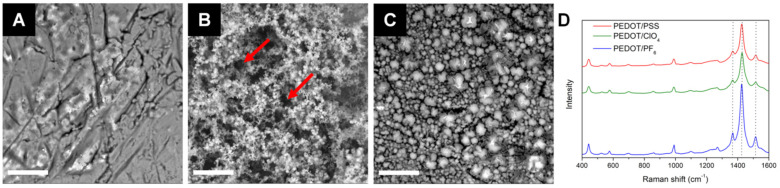
Surface characterization. SEM images obtained at 15 kV for (**A**) PEDOT/PSS, (**B**) PEDOT/PF_6_, and (**C**) PEDOT/ClO_4_; scale bar represents 30 µm. Red arrows indicate pores in the polymer structure. (**D**) Raman spectra of the polymer coatings collected over the range of 400–1600 cm^−1^.

**Figure 3 polymers-13-01948-f003:**
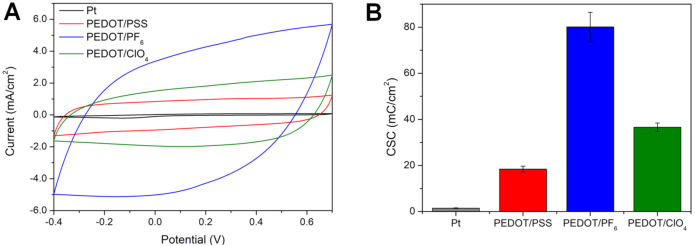
Cyclic voltammetric characterization of PEDOT matrices. (**A**) CV curves collected in 0.1 M KCl at a scan rate of 100 mV/s; (**B**) Comparison of CSC values calculated from the CV data for a bare platinum electrode as well as the electrodes coated with PEDOT/PSS, PEDOT/PF_6_, and PEDOT/ClO_4_; *n* = 3.

**Figure 4 polymers-13-01948-f004:**
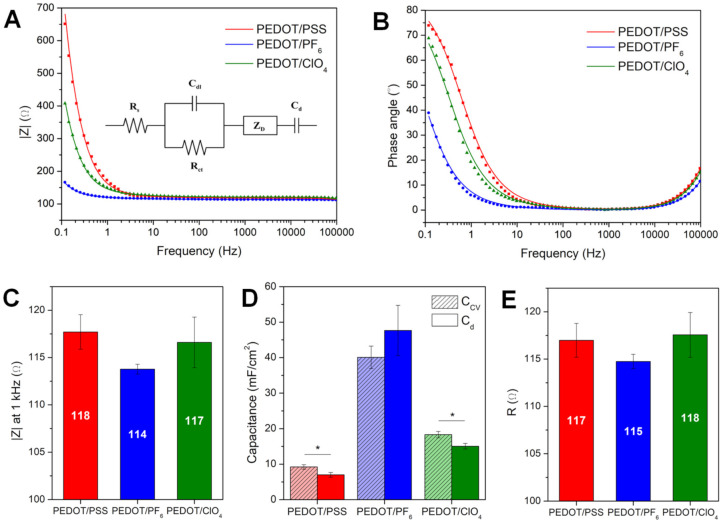
Impedance characterization of PEDOT matrices. EIS data in the form of Bode plots showing the frequency-dependent behavior of the (**A**) impedance modulus and (**B**) phase angle of platinum electrodes coated with PEDOT/PSS, PEDOT/PF_6_, and PEDOT/ClO_4_; the equivalent circuit is presented as an inset; dots represent experimental data and lines represent simulated results. (**C**) Magnitude of impedance at 1 kHz, (**D**) comparison between cyclic voltammetry capacitance (C_CV_) and polymer capacitance obtained from EIS fitting (C_d_), and (**E**) total resistance R (R_s_ + R_ct_) for PEDOT samples doped with PSS^−^, PF_6_^−^ and ClO_4_^−^; * *p* < 0.05; *n* = 3.

**Figure 5 polymers-13-01948-f005:**
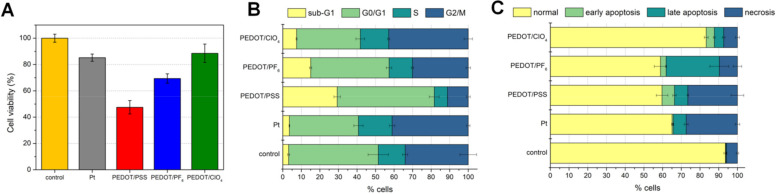
Biological characterization of PEDOT matrices. (**A**) Cell viability based on an MTT assay; (**B**) cell cycle distribution; (**C**) Annexin V and propidium iodide-based apoptosis assay for B35 cells cultured on investigated PEDOT materials, bare Pt electrode and control empty (uncoated) well, for 48 h.

**Figure 6 polymers-13-01948-f006:**
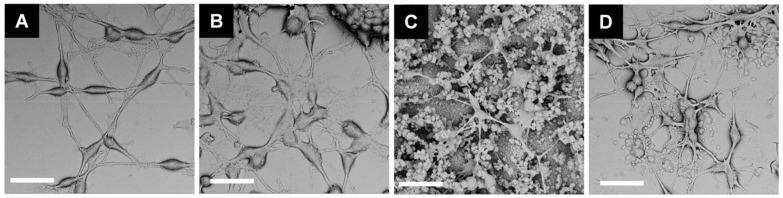
Morphology of B35 cells cultured on PEDOT matrices. SEM micrographs of B35 cells fixed after 48 h cultured on the surfaces of (**A**) platinum, (**B**) PEDOT/PSS, (**C**) PEDOT/PF_6_, and (**D**) PEDOT/ClO_4_. The scale bar represents 30 µm.

## Data Availability

The data presented in this study are available on request from the corresponding author.

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
