# Peer review of "Dopant-Dependent Electrical and Biological Functionality of PEDOT in Bioelectronics"

_polymers, 2021, doi:10.3390/polym13121948_

Round 1

Reviewer 1 Report

This study by Małgorzata Skorupa et al., reports on the fabrication and characterization of three PEDOT-based materials fabricated by electrochemical polymerization of EDOT in the presence of different counterions, namely PSS− (poly(styrenesulfonate)), ClO4− (perchlorate), and PF6− (hexafluorophosphate).

It is claimed that the type of dopant has a strong impact on several film properties, namely on the impedance, charge storage capacity, and biocompatibility.

The message of this work is relevant, and it shows that by doping is possible to optimize the PEDOT properties for bioelectronic applications.

The manuscript is very well-written and clear very well and it is quite pleasant to read. The conclusions are strongly supported by a carefully planned experimental work. It is my opinion this work should be published.

However, there are a few aspects in the manuscript that need to be clarified or better elaborated. Below I outline them:

  • On line 319 The text says: “The analysis of phase angle vs. frequency plots (Figure 4B) demonstrated notable variations in the relaxation frequency exhibited by the samples, suggesting distinct capacitance of the studied PEDOT films”.

Honestly, I cannot see a “relaxation frequency” on the plot of Figure 4B neither in Figure 4A.  I see that the impedance magnitude at low frequencies is different.  Metal/electrolytes often exhibit Maxwell-Wagner relaxation but is not the case of PEDOT based materials because of the low interfacial impedance often observed in PEDOT/electrolyte systems.

I think the authors mean that the impedance has a strong frequency dependence. This is what we see in the Figures 4A and 4B.  But I think the word relaxation does not really apply to the data represented on Figure 4. I recommend clarifying what it is meant by “relaxation frequency”.

  • In plate E of Figure 4 It is not clear what is he impedance parameter represented. It is stated that is the “total resistance”. But “total resistance” is not a parameter on the equivalent circuit (depicted on the plate A of Figura 4A).

Is the total resistance the low frequency interfacial resistance of the electrical double/layer resistance which is called contact resistance (Rct)? or it is the high frequency resistance (Rs)?

Or is just the resistance measured at 1 kHz?

If it is the high frequency resistance RS, then the authors have to bear in mind that this resistance is comprised by the series sum of the bulk polymer resistance and the electrolyte resistance.

The series bulk resistance of the polymer films is also a quite an interesting parameter for bioelectronic applications. The reason is as follows:

For signal coupling between the cell and the electrode the interfacial impedance of the electrical double-layer is crucial. However, the signal must be transported across the polymer track to the outside circuit. If this series bulk resistance of the polymer is high, it degrades the performance of the bioelectronic sensor. If the values of RS are extracted and presented will be an added value for this study.

In addition, accordingly to some recent studies, the resistance component of the impedance is crucial to achieve ultra-low noise PEDOT electrodes for bioelectrical signal detection. If this study clearly shows the effect of doping on the resistance it will be an important added value.

As a comment, I was expecting that doping should essentially affect the resistance, but apparently it is the capacitance that undergoes major changes upon doping. Perhaps the authors can speculate on this intriguing point.

  • The modified Randles circuit used in this work has unnecessary elements.

Why is a polymer capacitance included (Cd)?

 It is said that “The capacitive character of the polymer coating is represented by a capacitor element Cd in the equivalent circuit”.

The surface of the polymer charges forming and electrical double layer. Charges on the polymer bulk compensate ionic charges on the polymer/electrolyte interface surface. The capacitance that makes sense here is the electrical double-layer capacitance and possible the electrolyte capacitance. The concept of a polymer capacitance makes sense if the film is dried and behaves as a solid-state electrode (in my opinion).

Furthermore, if a Cd is added in series with the electrode, in practical terms it means that the circuit does not conduct for direct current signals (DC signals), and this is not true.

 In summary, I agree that the circuit fits the data, but it can be simplified.

Reviewer 2 Report

The authors submitted a manuscript titled “Dopant-dependent electrical
and biological functionality of PEDOT in bioelectronics”. In this work,
conducting films based on PEDOT matrix, with properties modulated
through doping PSS-, ClO4- and PF6-. It is displayed that through the
choice of a dopant and doping conditions, PEDOT-based materials can be
efficiently tuned with diversified physicochemical properties. I suggest
the manuscript to be accepted after some revisions. The followings are
my suggestions.

1. For broad impacts, other applications of conductive polymers need be
introduced and the following papers need be cited: Composite of Strip-shaped ZIF-67 with Polypyrrole: A Conductive Polymer-MOF Electrode System for Stable and High Specific Capacitance, Engineered Science, 2021, 13, 71-78; Polypyrrole Functionalized Graphene Oxide Accelerated Zinc Phosphate Coating Under Low-Temperature, ES Materials & Manufacturing, 2020, 9, 48-54
2. In the manuscript, a few details need to be revised. For example, μl or μL, etc.
3. The image resolution should be improved.
4. In the introduction, recent progress about PEDOT for biological use should be exhibited.
5. For the electrochemical part, further discussion is needed and the following papers should be cited: Chemically Deposited NiCo2O4 Thin Films for Electrochemical Study, ES Materials & Manufacturing, 2021, 11, 16-19; Hybrid solid state supercapacitors (HSSC’s) for high energy & power density: an overview, Engineered Science, 2020, 12, 38-51
6. The reference format needs to be unified. 

Author Response

The authors submitted a manuscript titled “Dopant-dependent electrical and biological functionality of PEDOT in bioelectronics”. In this work, conducting films based on PEDOT matrix, with properties modulated through doping PSS-, ClO4- and PF6-. It is displayed that through the choice of a dopant and doping conditions, PEDOT-based materials can be efficiently tuned with diversified physicochemical properties. I suggest the manuscript to be accepted after some revisions. The followings are my suggestions.

  1. For broad impacts, other applications of conductive polymers need be introduced and the following papers need be cited: Composite of Strip-shaped ZIF-67 with Polypyrrole: A Conductive Polymer-MOF Electrode System for Stable and High Specific Capacitance, Engineered Science, 2021, 13, 71-78; Polypyrrole Functionalized Graphene Oxide Accelerated Zinc Phosphate Coating Under Low-Temperature, ES Materials & Manufacturing, 2020, 9, 48-54

We would like to thank the Reviewer for the positive opinion on our work. According to the suggestion of the Reviewer, we have described additional applications of conducting polymers in the Introduction section (page 2).

  1. In the manuscript, a few details need to be revised. For example, μl or μL, etc.

The manuscript has been carefully checked, and few details have been corrected.

  1. The image resolution should be improved.

According to the suggestion of the Reviewer, the resolution of images has been improved (≥300 dpi).

  1. In the introduction, recent progress about PEDOT for biological use should be exhibited.

We would like to thank the Reviewer for this suggestion. Accordingly, we have added a section devoted to the recent progress in the design of PEDOT-based bioplatforms (Introduction, page 2).

  1. For the electrochemical part, further discussion is needed and the following papers should be cited: Chemically Deposited NiCo2O4 Thin Films for Electrochemical Study, ES Materials & Manufacturing, 2021, 11, 16-19; Hybrid solid state supercapacitors (HSSC’s) for high energy & power density: an overview, Engineered Science, 2020, 12, 38-51

As suggested by the Reviewer, we have supported the discussion of electrochemical results by citing recommended papers (page 9).

  1. The reference format needs to be unified. 

The references have been checked, unified and updated.

Round 2

Reviewer 2 Report

It can be recommended for publishing.